# Towards Sustainable Soil Stabilization in Peatlands: Secondary Raw Materials as an Alternative

Zane Vincevica-Gaile [1,*], Tonis Teppand [2], Mait Kriipsalu [2], Maris Krievans [3], Yahya Jani [4], Maris Klavins [1], Roy Hendroko Setyobudi [5], Inga Grinfelde [6], Vita Rudovica [7], Toomas Tamm [2], Merrit Shanskiy [8], Egle Saaremae [2], Ivar Zekker [9] and Juris Burlakovs [2]

1  Department of Environmental Science, University of Latvia, LV-1004 Riga, Latvia; maris.klavins@lu.lv
2  Chair of Rural Building and Water Management, Estonian University of Life Sciences, 51014 Tartu, Estonia; tonis.teppand@emu.ee (T.T.); mait.kriipsalu@emu.ee (M.K.); toomas.tamm@emu.ee (T.T.); egle.saaremae@emu.ee (E.S.); juris.burlakovs@emu.ee (J.B.)
3  Department of Geology, University of Latvia, LV-1004 Riga, Latvia; maris.krievans@lu.lv
4  Department of Urban Studies, Unit of Built Environment and Environmental Science, Malmö University, 211 19 Malmö, Sweden; yahya.jani@mau.se
5  Department of Agriculture Science, University of Muhammadiyah Malang, Malang 65145, Indonesia; roy_hendroko@hotmail.com
6  Scientific Laboratory of Forest and Water Resources, Latvia University of Life Sciences and Technologies, LV-3001 Jelgava, Latvia; inga.grinfelde@llu.lv
7  Department of Analytical Chemistry, University of Latvia, LV-1004 Riga, Latvia; vita.rudovica@lu.lv
8  Chair of Soil Science, Estonian University of Life Sciences, 51014 Tartu, Estonia; merrit.shanskiy@emu.ee
9  Institute of Chemistry, University of Tartu, 50411 Tartu, Estonia; ivar.zekker@ut.ee
*  Correspondence: zane.gaile@lu.lv

**Abstract:** Implementation of construction works on weak (e.g., compressible, collapsible, expansive) soils such as peatlands often is limited by logistics of equipment and shortage of available and applicable materials. If preloading or floating roads on geogrid reinforcement or piled embankments cannot be implemented, then soil stabilization is needed. Sustainable soil stabilization in an environmentally friendly way is recommended instead of applying known conventional methods such as pure cementing or excavation and a single replacement of soils. Substitution of conventional material (cement) and primary raw material (lime) with secondary raw material (waste and byproducts from industries) corresponds to the Sustainable Development Goals set by the United Nations, preserves resources, saves energy, and reduces greenhouse gas emissions. Besides traditional material usage, soil stabilization is achievable through various secondary raw materials (listed according to their groups and subgroups): 1. thermally treated waste products: 1.1. ashes from agriculture production; 1.2. ashes from energy production; 1.3. ashes from various manufacturing; 1.4. ashes from waste processing; 1.5. high carbon content pyrolysis products; 2. untreated waste and new products made from secondary raw materials: 2.1. waste from municipal waste biological treatment and landfills; 2.2. waste from industries; 3. new products made from secondary raw materials: 3.1. composite materials. Efficient solutions in environmental engineering may eliminate excessive amounts of waste and support innovation in the circular economy for sustainable future.

**Keywords:** circular economy approach; material cycling; road construction materials; soil stabilization; secondary raw materials; waste valorization; weak soil

## 1. Introduction

Peatlands cover over 4 million km² of the earth's area, primary providing globally significant ecological services and functions such as water storage and filtration, flood control, coal (peat) management, carbon storage and sequestration, transpiration cooling, habitat for wildlife, and recreational area. They generally involve bogs, swamps, marshes, mires, and wetlands [1–5]. Even though the most considerable areas are concentrated in

the northern regions (Figure 1), peatlands are widespread from subarctic to equatorial zones. However, not all peatlands may be precisely recognized due to regionally variable definitions and specifics of classification [4,6,7]. According to the Food and Agriculture Organization of the United Nations (FAO), peaty soils refer to Histosols [5,8].

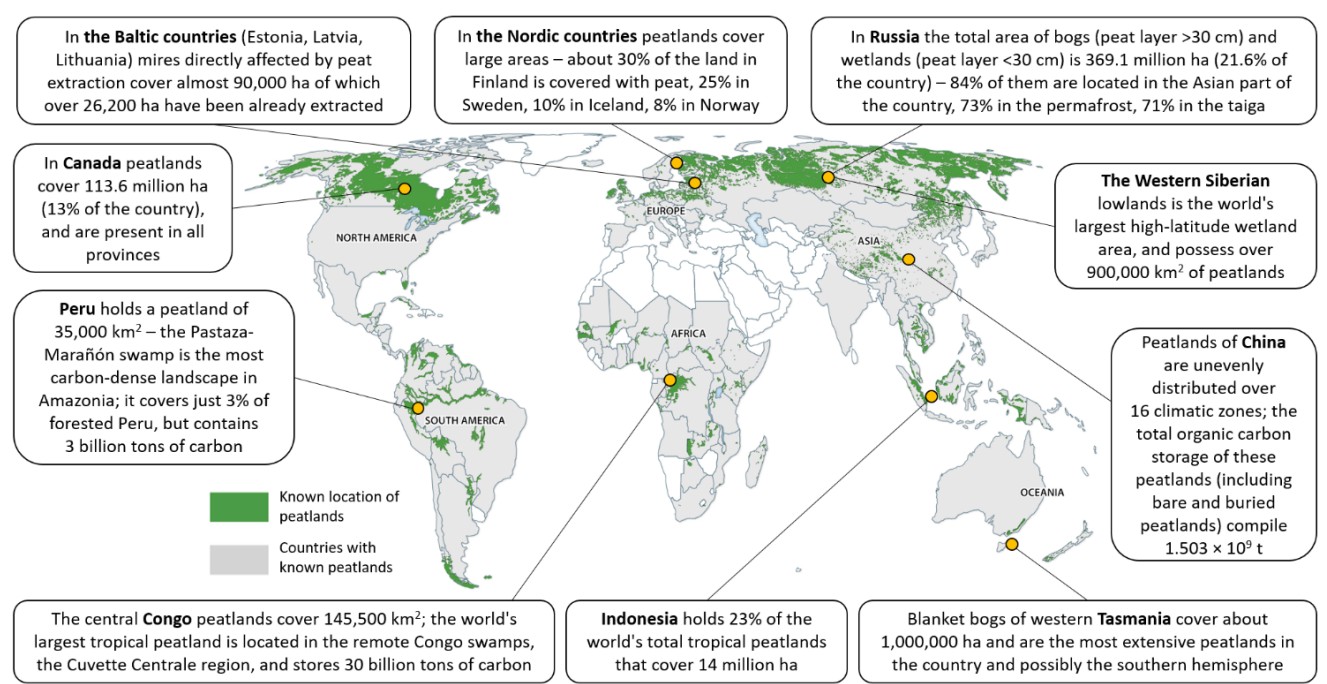

**Figure 1.** Distribution of peatlands worldwide [7] and complementary facts [9–18].

For centuries, peatlands remained as useless, low-value areas due to their geological and environmental specifics: weak and unstable soil basis, high moisture content, sparse vegetation, inappropriate environmental conditions for agriculture and settlements, and limited accessibility [19].

However, industrial development led to the urbanization of peatlands. For example, cities such as Saint Petersburg and Amsterdam and the airports of Kuala Lumpur and Zurich are built on peatlands. Presently, the growing world population and the need for land make peatlands more and more exposed to industrial activities. The location of vast areas of peatlands along rivers and near coastlines reinforces the need for infrastructure to develop ports, resorts and towns, and access infrastructure: roads, highways, railways, airports, power lines, etc. [1,20].

At an industrial scale, peatlands are exploited for fossil resource extraction (mainly of oil, gas, and peat) that cannot be carried out without appropriate transport, power, and maintenance infrastructure objects [1,21–23]. Undeniably, industrial activities are not in favour of the environmental protection and ecology of peatlands, but if they are inevitable, they can be provided in a more sustainable way than in decades before [24,25].

Peatlands in equatorial and tropical regions are affected by high temperature and humidity in combination with intense precipitation and soil erosion. In contrast, those located in subarctic regions are exposed to the specific impacts of seasonal frost variations, natural permafrost conditions, and permafrost deformation induced by climate change [6,26]. Site-specific climatic and geological circumstances require modifications and modernization of existing conventional methods and technologies applied for soil stabilization, including searching for new materials [27]. Global economic growth demands better durable and cost-effective infrastructure constructions, mainly due to the necessities related to mining and forestry. Therefore, innovative sustainable soil stabilization approaches that integrate circular economy principles are needed. In the best case scenario, the circular

economy requires any produced waste (either of domestic, municipal, or industrial origin) to be turned back into another valuable applicable for the economy as a secondary raw material [28–31]. General management of such materials is similar to that of primary raw materials, i.e., the trading and transportation regulations in the European Union (EU) are the same; however, changes in waste-related legislation are needed to stimulate sorting, recycling, and reuse of materials. While the quality standards for secondary raw materials are still under development, it is complicated to use these materials on a broad scale [28]; however, it is only a question of time, as the scarcity of resources is pressing on the minds of society and the wallets of industry stakeholders. Valorization of secondary waste materials corresponds to the Sustainable Development Goals set by the United Nations and Circular Economy Action Plan. Developing innovative composite materials and analysing potential valorization opportunities of secondary materials, such as industrially produced ashes or other waste, into value-added products used in stabilizing soils means extending the life cycle of primary source materials, thus closing the loop in material recycling [32].

This paper aims to review the opportunities for secondary raw materials to be applied in geotechnical stabilization of peatlands as an alternative to conventional materials considering scientific studies published from 2000 to the present. The goal is to understand technical aspects and to overview sustainable materials supporting the circular economy approach in weak soil stabilization.

## 2. Challenges in Peatland Stabilization

### 2.1. Demand for Stabilization

The process of infrastructure construction outside the zones of stable soils with good bearing capacity is a challenge for engineers, and searching for new technologies and materials is appreciable [33]. On weak soils, if preloading or floating roads on geogrid reinforcement or piled embankments cannot be implemented, then soil stabilization is needed; otherwise, safe engineering planning, including the construction of vitally necessary infrastructure, is practically impossible [27,34,35]. Several reasons emphasize the importance of peatland stabilization, i.e., peaty soils generally inhere harsh geotechnical conditions due to the low value of undrained shear strength at natural conditions, high water content, and low permeability, as well as low strength in combination with high compressibility and shrinkage on drying [35–38]. Focusing on the compressibility, peaty or boggy areas are characterized by '*uncontrolled or unexpected decomposition in fibrous peat deposits that may cause significant additional settlement of bearing strata, adversely impacting on the performance of engineering structures founded on or within such deposits*' [39]. Investigation of decomposition rates has revealed that organic matter content has the main influence on the mechanical properties of peat [40,41]. Although compression may increase the strength and reduce the compressibility of peat, loss of water can become a problem when peat, after the irreversible process of dry-out, may sorb back only an insignificant part of moisture in comparison to the initial amount [38].

Implementation of soil stabilization means change and improvement of soil physical properties in favour of engineering purposes; however, it may also influence the chemical properties of soil and surrounding waters [35,42].

### 2.2. General Principles of Stabilization

In road construction, it is a common practice to form structured layers with large bearing capacity. A foundation with nondeformable properties is advisable, and, from Roman times up to contemporaneity, the easiest way to stabilize soil is mixing it with 'pozzolana' (aluminium oxide and silica from volcanic ash) combined with lime [43,44]. From the 1920s, when hydrated lime was applied for soil stabilization, the era of modern stabilization started, first in the USA, then in other regions of the world [44]. The soil type and mineralogy determine combinations and specifics of a binder material. Spreading, mixing, and compaction of soil with the binder are essential in the presence of water. The process is run by cation exchange (exchangeable cations are replaced by higher valence

calcium ions of lime), agglomeration (larger effective grain sizes), and pozzolanic reaction (cementitious material forms) [45–48].

A range of methods have been developed and exploited (Figure 2) for general soil improvement, starting with the basic ones, such as soil excavation and replacement and drainage, followed by advanced ones, such as soil stabilization by adding various materials, chemical stabilization, variable modification (mechanical, hydraulic, electrical, thermal), the reinforcement method, etc. [27,35,49].

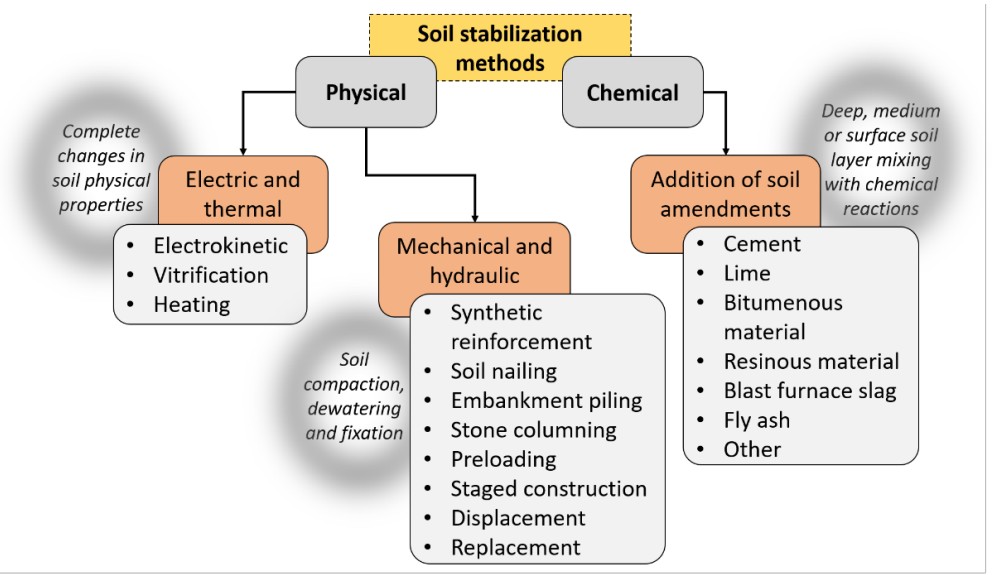

**Figure 2.** General soil stabilization methods and their impact on soil [27,35,50–55].

Depending on the site specifics and geotechnical needs, the methods are applicable for shallow, medium, or deep soil stabilization, and generally are categorized into two types: a) mechanical approaches involving displacement or replacement of soil, staged constructions, preloading, stone columns method, embankment piling and synthetic reinforcement applications [52]; and b) chemical approaches through deep mixing and surface stabilizations in situ by using amendments such as cement, fly ash, bottom ash, bentonite, gypsum, silica fume, blast furnace slag, etc. [53–55].

As an example of a purely mechanical approach using a secondary raw material, a case study (DREDGDIKES project) at the Vistula River (in Poland) can be mentioned, where the dike body was constructed from a composite material containing 70% coal ash in a mixture with 30% dredged sand derived from the same river. The study reveals that bottom coal ashes are characterized by right granularity, pozzolanic and hydraulic properties, and high shear strength [56]. Soil–ash composites are relatively cheap materials, and for geotechnical stabilization of soil, these materials compete with classical materials such as cement [57]. Production of cement emits significant amounts of $CO_2$, but the amount of cement is significantly reduced in the composition of binder, using instead secondary raw (recycled) materials (e.g., thermal treatment products of waste such as ashes) [58].

However, the most common practice for soil stabilization involves combining mechanical methods with chemical stabilization, even if chemical processes occur passively [27].

### 2.3. The Problem of Permafrost

Peatlands located in subarctic regions, such as north of Europe and Asia, Alaska, Canada, and islands of the Arctic Ocean, are exposed to the specific impact of permafrost; furthermore, in the circumpolar permafrost region, almost 20% of the terrain is peatland [23]. In permafrost regions, often the uppermost layer (the active zone) is subjected to freeze and thaw, but the permafrost is below the active layer. At permafrost conditions, soil temperature usually does not exceed 0 °C for several years, but if changes in the environ-

ment occur and the climate becomes warmer, then the physical and mechanical strength of soil reduces irreversibly [26]. Ice and voids, fine soil, and mineral and organic matter in undisturbed permafrost have attained the equilibrium state over time. However, upon any engineering construction of infrastructure, this equilibrium changes, and permafrost deformation occurs (Figure 3a). For example, removing vegetation leads to faster thawing of permafrost; constructed embankments with lower albedo transmit more heat to the underlying permafrost [26,59–61]. These are the challenges in permafrost that require specific construction management (Figure 3b) if mass stabilization is planned. Several studies emphasize that increasing ambient temperature accelerates permafrost degradation, and due that these regions are most vulnerable [60,62], and that nothing indicates that the situation can change.

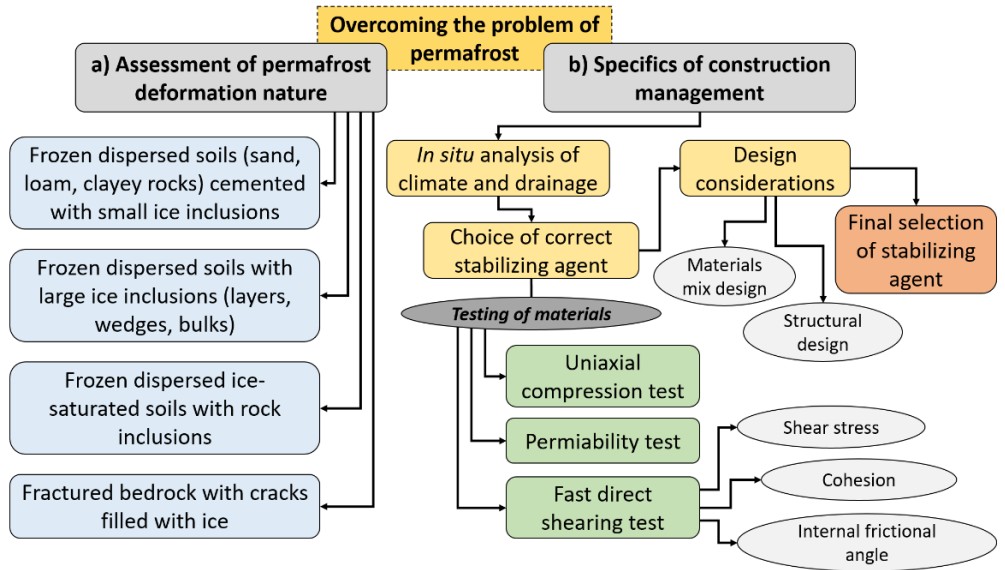

**Figure 3.** Permafrost deformation nature (**a**) and its construction management specifics at stabilization (**b**) [23,26,60,63–66].

### 2.4. Mass Stabilization as an Option

Already for decades, mass stabilization technology (sometimes called a Nordic approach) has been intensively used in Fennoscandia and North America. It can be applied for routine soil stabilization as well as in specific conditions such as permafrost areas, which are very common in peatlands of certain latitudes [67]. This technology is used to improve the soil bearing capacity under railways, roads, and warehouses. Furthermore, mass stabilization is also functional for mechanical and chemical binding of pollutants if such persist in former industrial or otherwise heavily anthropogenically influenced areas [68].

The Nordic approach in peat stabilization, according to the guidelines, is based on cement mixture with furnace slag and/or gypsum. In the Baltics, cement and oil shale ash have proven to be an effective binder in soil mixtures [69]. Fibrous peat more easily surrenders to the stabilization process than amorphous peat, and it is advisable to mix the treated peat layer with various subcomponents such as sand or clay in addition to the binder materials [70,71]. The first step of stabilization is prehomogenization that involves adding the components into the soil, further elaborating the final result of mass stabilization [72]. Stabilization provides additional challenges, where detailed studies of design, climate, and drainage require more comprehensive analysis [73,74].

Mass stabilization as a soil improvement method applicable for weak soils, including permafrost regions, has proven itself in practice for more than 25 years in more than 30 countries. Therefore, the technical background of mass stabilization is developed at a high level. The equipment is relatively light and mobile, and thus easily transportable and

applicable at various locations [75]. A specific way of mass stabilization (Figure 4) at a particular location has to be set after a detailed geological, climatic, and other assessment. For example, mass stabilization in the form of triangle piles consumes fewer materials, while other options can involve block-form or column-form stabilization options [58].

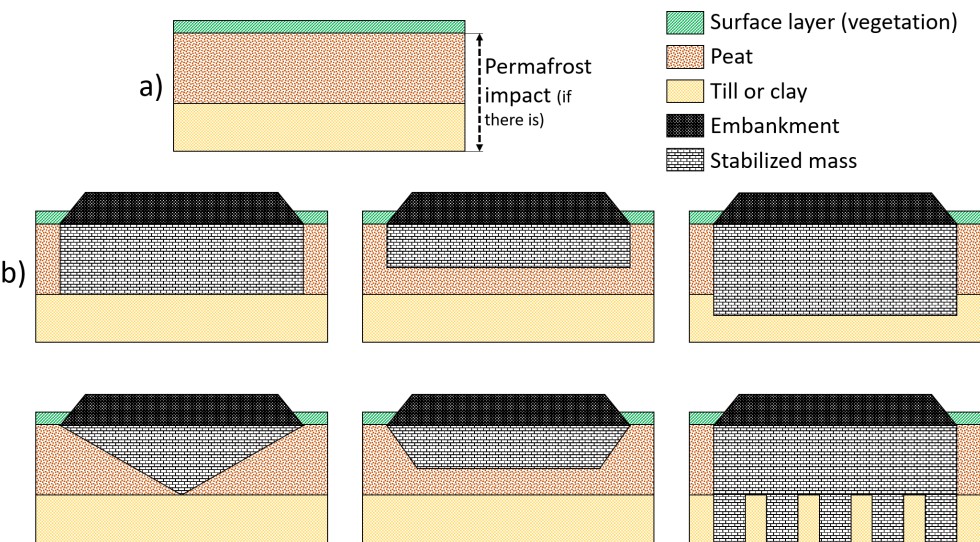

**Figure 4.** Schematic cross-sections of approach for mass stabilization in peatlands—nonstabilized peatland (**a**) and variations of stabilization (**b**) [58,72,76].

From the geotechnical standpoint, mass stabilization is forced by the hydration process after the binder (e.g., fly ash or slag) is mixed into the soil. Long-term studies reveal that the strength of stabilized soils improved by 1.6 times, and the process is cost-effective in areas where transportation possibilities for excavated peat and availability of construction materials for soil replacement are limited [58,77]. A study [78] indicates that about 70% of the unit price of mass stabilization can depend on the price of a binder. Conventionally used cement has a considerably high carbon footprint and is expensive. The replacement of it with binders based on secondary raw materials, such as industrial waste (e.g., oil shale ash, fly ash, furnace slag powder, gypsum, etc.), was tested both in a laboratory and in the field. The results of the leaching tests were in favour of the use of fly ash-based binders in the mass stabilization of peatlands [72,79,80].

## 3. Applying a Circular Economy Approach in Material Choice for Stabilization

Growing environmental concerns, governmental support, and changes in thinking motivate people to focus on more sustainable businesses and to promote new regulations incorporating the principles of sustainability, especially in material recycling [29,31]. The primary efforts should be devoted to waste reduction and long-term value creation to materials where the critical role is attributed to designing for (a) recycling; (b) remanufacturing and reuse; (c) disassembly and sorting; and d) the environment [31]. Stronger regulations on pollution and energy efficiency are working as a driving force for faster production changes towards the compromise between the goals of the environment and economics [81]. Substitution of primary raw materials with secondary ones may significantly diminish the need for virgin raw materials usually obtainable from ore mining. Application of various waste materials such as fly ash derived from the energy industry, palm oil fuel ash, natural fibres, alum waste, etc., for soil stabilization have been tested and described in the literature. The approach based on circular economy principles requires eco-efficient and economical solutions in geotechnical engineering. However, the choice on-site should be based on analysing the industrial loop and accessibility of secondary raw materials from local industries. An overview of the studies of the latest decades provides information on a

wide range of secondary raw materials, apart from conventional materials, applicable for soil stabilization. The main groups of secondary raw materials include thermally treated waste materials and industrial waste, as listed in Appendix A Regarding the terminology, some waste materials can be categorized as byproducts. However, due to specific criteria set by legislation in the EU that have to be fulfilled [82], this article avoids referring to the term 'byproduct'; thus, all secondary raw materials are assessed as a waste focusing on life cycle extension. European governments, as well as multilateral bodies in Europe, are increasingly interested in the strategic utilization of secondary raw materials to reduce produced amounts of waste [83].

It is evident that various constituents are used for soil stabilization as a binder or amendment material depending on the region's economic activity. However, the application of secondary raw materials should become incorporated in the routine management of infrastructure development [27,34]. Some of the potential amendments for soil stabilization are produced by industries in vast amounts. Therefore, an option to choose additional material recycling instead of waste disposal is paramount, because the target is to diminish the hazardous impact of waste masses and simultaneously reduce the use of primary raw material (cement, lime, sand) that demands additional efforts of mining and production activities [84].

Conventional approaches for peatland stabilization are always applicable due to developed and proofed technologies; however, beneficial management recommends saving of primary resources and reduction of greenhouse gas (GHG) emissions during the process of construction and building [85]. Studies have led to attempts using a relatively wide choice of raw materials acquired through mining and local resource extraction. However, an option to exploit secondary raw materials (various waste materials and byproducts) is promising due to broad availability and low costs [50,51]. One approach involves soil stabilization with waste materials such as shredded particular solid waste or fractions of municipal waste, turning them into valuable materials applicable for infrastructure development [86]. Other materials are geopolymeric binders used as an alternative to standard Portland cement for building materials. Several solid waste types, including combustion ashes, metallurgical slag, or glass processing waste, might become sources for construction materials with lower carbon footprint and energy consumption [87]. Alkali-activated materials are mainly varieties of blast furnace slag with low calcium content and coal fly ash; these exhibit properties necessary for specific applications such as high compressive strength, low thermal conductivity, acid and/or fire resistance, etc. While fly ashes of class F with low calcium content are most commonly used to synthesise geopolymeric materials, the use of fly ashes of class C with high calcium content has also been evaluated [88–92].

Valorization of various solid industrial wastes through alkali activation is a promising method for the reuse of particular disposed residual [93–95]. Through the addition of an alkali source (activator), the primary aluminosilicate structure of the material is modified to produce secondary polymeric silicate binder phases that improve the cementitious properties of the subject material. Mortar and concrete, based on alkali-activated materials, can provide mechanical performance similar to ordinary Portland cement systems, can have high fire and acid resistance, and can be used to stabilize/solidify other wastes, including immobilizing heavy metals [96].

## 4. Peculiarities Regarding the Use of Certain Secondary Raw Materials

### 4.1. Solving the Problem of Oil Shale Waste

Fossil fuels of low quality (including oil shale) are found worldwide, but their usage at an industrial scale is limited to few countries, e.g., Brazil, China, Estonia, Jordan, and Israel [97]. The main reason is related to the low calorific values of such fuels, as well as technological and environmental issues. For decades, the oil shale industry has produced much waste, mainly oil shale ashes, resulting, for example, in huge white mountains located in northern Estonia affecting the surrounding environment through possible hazardous

leaching. Kukersite oil shale has been mined for nearly 100 years, making Estonia the biggest producer of solid oil shale in the world [97–99]. Calorific values of this fuel are relatively low, and production creates 40–50 wt% of solid waste (Figure 5), the majority (~98%) of which is landfilled [97]. Moreover, processing leaves ~45–50% of solid waste (oil shale ashes), which yet has limited secondary usage. Several problems accompany oil shale application as a fuel at power plants, including an enormous amount of alkaline ashes, a leftover at power plants from oil shale burning, deposited nearby [100]. More than 7 Mt of ashes are produced yearly by thermal power plants in Estonia [101,102].

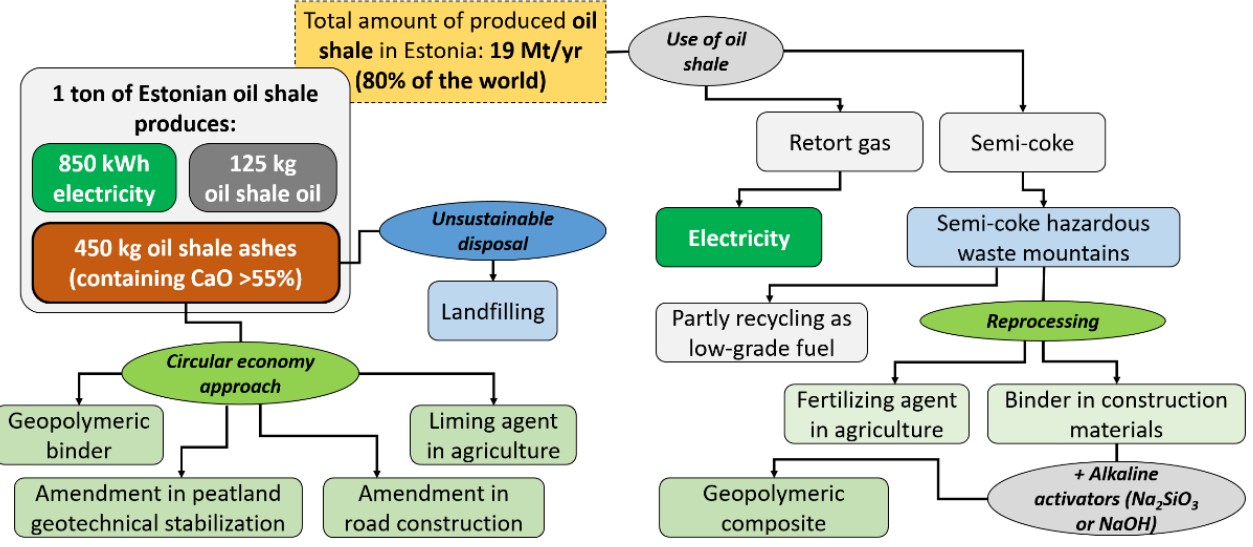

**Figure 5.** Processing of oil shale, indicating the applicability of the circular economy approach [101–103].

Oil shale comprises organic, carbonate, and terrigenous material at various proportions, but high calcium content (CaO content as high as 55%) guides for valuable utilization of this waste [101]. Its consistency is similar to fly ashes derived from coal combustion and is applicable for road construction and agricultural purposes; however, 95% of the amount is deposited in large ash fields next to power plants. A fraction with higher concentrations of $SiO_2$ and $Al_2O_3$, compared with other solid waste resulting from oil shale processing can be used as an additive in ordinary Portland cement. However, other fractions of ashes are exposed to hydraulic transportation in a slurry with a water-to-ash ratio of 20:1 in sedimentation ponds, creating mountain plateaus in Estonia occupying an area of more than 20 km$^2$ and containing more than 300 Mt of ashes [104–106]. The physical, chemical, and mineralogical properties of oil shale ashes are quite widely studied [107,108], but secondary use is still limited. The geopolymeric potential of oil shale ashes for construction applications and alkaline activation under ambient conditions has been investigated by analytical techniques allowing identification of formation of amorphous C-(A)-S-H gel, giving the samples eight times higher compressive strength (8.5 MPa) than for samples exhibiting cementitious properties upon hydration (1.25 MPa) [109]. Trace element concentration in oil shale ashes is acceptable for industrial areas except for arsenic detectable in ashes derived from electrostatic precipitators (containing about 40 mg/kg of arsenic in comparison to the limit value of 20–30 mg/kg for residential areas and 50 mg/kg for industrial zones). However, ashes from precipitators constitute only 23% of all ash amount generated [110]. The primary application of oil shale ashes still involves amending as additives in Portland cement, filter material for wastewater treatment, and as a soil liming agent in agriculture [108,111]. However, the cementing bonds developed upon simple hydration of Ca-rich oil shale are weak and chemically unstable under continuing leaching and aggressive environments [109,112,113], making the waste unsuitable for long-term construction and/or stabilization applications. Single oil shale usage, specifically for oil and

gas production, is projected to rise globally, and it is essential to improve the cementitious properties of waste to develop new sustainable and environmentally less harmful usages of this waste [50]. Instead, oil shale fly ashes have appropriate properties to be used combined with other organic matter such as peat. Advanced construction material is environmentally sustainable if it contains only local materials and waste as raw materials. Peat is a partially renewable resource, continuing to accumulate on 60% of global peatlands; global peat reserves are about 12 billion tons in an estimated area of peatlands (about 400 million ha), whereas the extraction in 2014 was about 30 million tons [114]. Oil shale ashes or similar ashes from coal or lignite power plants are available in several regions worldwide. A combination of peat organic matter and oil shale ashes provide excellent properties of thermal resistance, bending stress, and compressive strength (around 1.2 MPa) thus can be used as a construction material [115]. Nevertheless, it is recommended to monitor the concentration of trace elements and leaching parameters in vicinity of surface waters if oil shale ashes are used in road construction [116]. However, surface water discharges require comprehensive calculations and modelling efforts [117]. Properties and environmental risks of oil shale ashes are summarized in Table 1.

**Table 1.** Recent studies on physical/chemical properties/processes and related environmental risks of oil shale ash application.

| Property/Process | Risk | Reference |
|---|---|---|
| Excessive content of chemical elements and constituents | Soil and groundwater pollution with arsenic, sulphates, barium | [107,108,112,118–122] |
| Content of lime | Changes in soil pH | [107,108,112,119] |
| Leaching | Groundwater and surface water contamination | [118,121–124] |
| Mobility and soil–water interaction | Groundwater and surface water contamination | [117,125–128] |

A case study on oil shale ashes used for road construction has been performed in a swamp area in eastern Estonia, and still, the road construction has affected the concentration of sulphates and barium in surrounding surface water. Geotechnical challenges are also significant; thus, new pilot studies for testing various structural design schemes are being done in wetlands [3] as to whether oil shale ashes as a stabilization material might be used in smaller amounts by varying the design of stabilized masses [35,50].

It might be even more stable to use the material reduction scheme, though not worsening the bearing stability parameters, and as less material of amendment is needed, this approach would result in qualitatively reduced leaching. Such tests are being done in Estonian peatlands for new forestry road construction using peat, fly ashes of oil shale, and water without any cement addition [103,116,119].

Oil shale semicoke is another hazardous waste produced by the oil shale retorting process. The global petroleum industry is now making a historic leap from conventional to unconventional oil and gas [129], and as a waste, oil shale semicoke causes significant hazards to the environment [130,131]. Utilization of semicoke in a road base or subbase course construction material would provide a great solution to the disposal problem of long-term accumulated oil shale semicoke in landfill sites [132]. A partial solution for disposing of semicoke is recycling it to some extent to burn it as a low-grade fuel, and another common utilization of semicoke is to reprocess it as a raw material for fertilizers and construction materials [133,134]. The active ingredient in semicoke may benefit the pozzolanic reaction with cement, allowing semicoke to have potential applications as an aggregate of cement-stabilized road base or subbase course construction, saving natural gravel resource and transportation costs [119,132]. The performance of various tests (raw material test, modified compaction test, unconfined compressive strength test, splitting tensile strength test, compressive resilient modulus test, and freezing–thawing test) has already proven the suitability of oil shale semicoke to be used as an amendment for subbase course material of a highway or a base course material of a low-grade highway [132].

### 4.2. Applicability of Municipal Solid Waste Bottom Ashes in Weak Soil Stabilization

Incineration of municipal solid waste (MSW) creates a large amount of combustion residuals, a significant part of which are bottom ashes (about 20% of the initial mass). In the USA alone, over 30 million tons of MSW was incinerated in 2014 (13% of the total domestic MSW) [135], which correlates to 5 million tons of residual bottom ash. Primarily, bottom ashes are combined with fly ashes in order to avoid classification as hazardous waste [136,137]. Unlike fly ashes, bottom ashes are still mainly disposed instead of being used as secondary raw material. Comprehensive research has been performed to identify beneficial applications for MSW bottom ashes, primarily applicable as secondary raw material in infrastructure [138–142]. Recycling of bottom ashes as an aggregate material in roadway construction has been carried out in Europe [143] and Asia [144] and is slowly gaining attention in the USA [145]. In some EU countries, recycling bottom ashes on roads is not only a popular approach but mandated by legislation in many cases [146]. MSW bottom ashes exhibit similar mineralogy and physical properties to some natural aggregates [147,148] and thus may prove economic advantages if traditional aggregates can be supplemented or replaced in roadways (especially when factoring in the avoided cost of landfilling). Furthermore, the utilization of recycled aggregate sources in road base decreases dependence on naturally mined minerals and offsets GHG emissions associated with the mining of virgin minerals [149,150]. In the USA, MSW bottom ash is predominantly classified as a nonhazardous waste by the Toxicity Characteristic Leaching Procedure. Also, studies have demonstrated that MSW bottom ash contains small amounts of metals and salts that may pose a risk to environmental and ecological health [151]. The risk posed by direct human contact with this material is low if it is placed beneath a compacted pavement layer; however, the leaching of contaminants to groundwater must still be considered. The idea of blending incinerator-derived bottom ashes with conventional granular materials (but not diluting hazardous waste) for road base has potential in the future, and the legislative aspects must be considered. However, the leaching of trace elements may not always be directly proportional to the original mass of ash present. The results support that blending MSW bottom ashes with natural and recycled aggregates may serve as a practical engineering control to mitigate environmentally leached and total concentrations of potentially toxic and harmful elements [152,153]. Since ash quantities are relatively small compared to aggregate demands in roadway construction, blending may create opportunities for ash recycling that were previously not feasible. Direct exposure risks appear to approach those posed by natural aggregates when blended to levels of 15%. Actual leaching risk evaluation requires site-specific conditions as well as road configuration and design knowledge to estimate likely concentrations at the compliance point. Recycled product in a financial matter outweighs the difficulties incurred [154]. The use of bottom ashes from MSW incineration as an amendment is prospective for peatland forestry road building in countries where waste incineration is a developed business (e.g., Sweden, Austria, Italy, Estonia) [35,103,139,142]. However, the EU is willing to take more action to promote the Circular Economy Action Plan's goals, including the reuse of secondary raw materials instead of waste incineration.

### 4.3. Perspectives of Fly Ashes from Power Industry for Forest Road Constructions

Power and heating plants generate a significant amount of fly ashes (i.e., ashes composed of fine particles) from the burning of biomass such as wood and peat, and, for example, in Finland in 2017, fly ash production was about 700,000 t [155,156]. Wood ashes have traditionally been utilized as a fertilizer for forest and agricultural soils [157,158]; other utilization possibilities involve their use as a binding material in cement [159,160] or road construction material [161,162] or in new materials in combination with organic-rich freshwater sediments [163]. Road construction materials, especially for forest and peatland roads, which have low daily traffic, are paved with gravel or even surfacing with local soil materials and have less demanding design specifications [164]. Asphalt roads are not under the influence of water that drains through the road structure, causing the leaching

of heavy metals. The utilization of biomass ashes has not yet been widely studied on forest roads, although some studies exist [165]. Bohrn and Stampfer [166] have stated that ashes improved the bearing capacity compared to a non-ash road structure. Another study performed by Kaakkurivaara and Korpunen [167] in a cost calculation model proved that ash structures are suitable options for the rehabilitation of forest roads. Research [128] involving a two-year survey on various parameters of wood-based and fluidized bed ashes has concluded that the chemical and physical state of ashes at application is under environmentally acceptable measures. Nordmark et al. [168], during a three-year survey, found that biomass ashes can be used without threatening the surrounding environment. Leaching studies, however, did not include the evaluation of peat-based ashes, where, for instance, higher concentrations of arsenic can be present [169]. The abundance of heavy metals in ashes from power plants is determined by the burning techniques and fuels used, the purity of the fuels, the combustion temperature, and the gas filtering equipment used [170]. The utilization of biomass ashes in the infrastructure of forest roads is estimated as environmentally safe, but for environmental safety purposes, consideration should be attributed to the possible leaching of heavy metals. A risk for leaching is higher where high concentrations of biomass ashes are widely spread in the road and where water from these parts is flowing directly to drainage ditches [156].

### 4.4. Application of Ashes from Other Industries in Peatland Stabilization

The pulp and paper industries generate a lot of ashes and seek efficient mechanisms and management for ecological and economical solutions in waste utilization [171]. Both laboratory and field experiments have been performed to assess the performance of pulp and paper ashes from technical, environmental, and economic perspectives in a variety of studies [172,173]. Field monitoring has revealed elaborative patterns for stiffness and lower permeability of ashes over time due to the hydration. The self-cementitious property of fly ashes from the pulp and paper industry arises from high calcium and aluminosilicates and plays a significant role in the improvement of strength and volume change properties for foundation soils [174–176]. Thus, ashes can be applied as a stabilizer for improving the engineering performance of expansive foundation soils, subgrades and subbase layers of roads, hydraulic layers, and similar constructions [177]. The permeability of such ashes is low, and they can be used as hydraulic barriers, including applications for landfill covering [178] and sustainable substitution of conventionally used materials [27,35]. This enables the conservation of natural resources by reducing energy demand, diminishing carbon footprint, and replacing materials such as cement in the building and construction industry. Broader utilization of ashes can also save valuable landfill space for other nonbeneficial waste materials [171]. The utilization of ashes in earth structures of transport engineering is one of the most important ways to decrease the volume of ash deposited in tailing dams [179].

Waste from wastewater treatment is formed from sewage sludge ashes that are left after the incineration of sewage sludge. It can be used as sustainable construction material if harmful heavy metals are stabilized and leaching avoided. Lightweight aggregate production is possible for concrete, blocks, mortar, road pavements, geotechnical stabilization material, and ceramics. Newly developed material must avoid unfavourable conditions such as acidic and heavy-moisture environments [180,181]. In themselves, sewage sludge ashes lack good binder properties, but they are applicable as asphalt filler. Different leaching criteria for analysis have been applied in various experimental works to draw comparisons impacting drinking water, soil quality, limits of hazardous material, and abrasion tests, which for environmental safety reasons are compulsory. Sewage sludge ashes are comparable with crushed concrete debris and cement kiln dust, thus restricting their use [182–184].

Palm oil fuel ashes (POFA) are waste from tropical industries in southeast Asian countries, e.g., Malaysia, and vast amounts of this material are generated annually [185,186]. The abundance of this solid waste inspired research to discover the properties and potential

uses of POFA; the material is classified as a pozzolanic and might substitute cement [187], aerated concrete [188], and lightweight aggregate concrete [189]. The strength of POFA specimens with crushed cockle shell is high, as an undisturbed hydration process for continuous generation of calcium silicate hydrate (C-S-H) gel filling the voids results in denser and stronger brick [190].

The increasing plantation areas of palm oil trees in Malaysia, Indonesia, and Thailand lead to continually rising POFA production [191]. As POFA, assessed as a material having no market value, is simply dumped into ponds or lagoons, it poses a serious environmental concern adversely affecting peatland sustainability. However, recent studies have shown that POFA is rich in silica and might be beneficial as a cost-effective and durable additive in mortars in the construction and road stabilizing sectors [192,193]. The synergy between POFA and ground blast furnace slags (GBFS) has contributed to solving the problem of rapid setting time for GBFS, providing a new type of green mortar [194]. POFA is a prospective material to be used in soil stabilization in peatlands of tropical regions.

### 4.5. Geopolymer Composites from Thermally Treated Waste

Geopolymer composites have been investigated as a possible substitute for conventional concrete over the past 15 years. The binding materials consist mainly of pozzolanic materials and alkaline activators and can be employed to construct secondary and structural elements of good strength and excellent durability [88–92]. Geopolymers from Jordanian oil shale ashes were investigated by Haddad et al. [195], revealing mechanical and durability properties of such materials at various processing procedures. It has to be considered that geopolymer composites need the addition of alkaline activator, such as sodium hydroxide, sodium silicate, or others, thus turning the process into a chemical reaction. Hence, several attempts were made to utilize such waste material to construct pavements and to produce cheap concrete. Findings have stipulated that oil shale ashes possess excellent pozzolanic characteristics and hence can be used as a base material in the production of geopolymer composites for masonry and structural applications [196,197].

The selection of raw materials and their processing conditions determine the properties of the products formed through alkali activation. Besides various slags and waste from glass manufacturing, combustion fly ash can also be useful for the production of geopolymer composites. Also, rice hull and biofuel ashes have been successfully used to produce alkali-activated materials using various alkali activators [198–200].

In most cases, the importance of the Si–Al ratio in raw material, the water–binder ratio, and the rates of the release of Si and Al from source materials have been found to have the most significant effect on geopolymer formation and properties [201]. Solid fossil fuel residues, commonly fly ashes of class F with low calcium content, are used for geopolymeric binder production, where the main reaction product is a three-dimensional crosslinked aluminosilicate network [92]. However, ashes of class C, i.e., fly ashes with high content of calcium, have also been successfully tested for alkali-activated and geopolymeric binders [88,89]. In this contribution, a geopolymeric potential of oil shale ashes reaching 6-7 MPa was obtained [109,111]; thus, the potential of using these ashes in peatland-crossing forest roads is highly prospective.

### 4.6. Some Other Specific Materials Used in Peatland Soil Stabilization

Natural plant fibres are frequent agricultural waste. Recently, natural fibre materials have gained momentum as an emulating soil-reinforcement technique in sustainable geotechnics [202]. The problems of these materials are related to hydrophilic and biodegradation properties, which create limitations in geotechnical engineering practices. Understanding the behaviour of the material at various subsoil conditions is essential for reliable stabilizations. The shrink–swell behaviour of fibres at subsoil conditions induces slipping and failures [203,204]. Thus, the current intentions of engineers target modifying existing ground improvement techniques by adding the unique advantages of natural fibre application, providing environmental friendliness, resource abundance,

minimal energy consumption, cost-effectiveness, and high potential over other established materials [205,206]. Bamboo, jute, coir, palm, sugar cane bagasse, and other fibres have high potential to stabilize soil in tropical countries where industry generates vast amounts of natural fibre secondary waste; tropical peatland stabilization would be one of the solutions [203].

Alum sludge is a waste generated by water purification and wastewater treatment plants when aluminium salts are used during the coagulation process [207]. Wet sludge composes up to 5% of the total quantity of processed water, and it is difficult and expensive to treat. Wet sludge, after dewatering and drying, is primarily disposed of in landfills [208]. The process of soil stabilization with alum sludge provides a low-cost soil stabilizer and gives solutions to the problem of waste management at a large scale [34,209]. Alum is the class of aluminium hydroxide minerals—a component of bauxite, the principal ore of aluminium. Generally, it is used to treat wastewater, and it is challenging to dump [34,210]. The addition of alum sludge to soil improves foundation strength indicated by California bearing ratio (CBR). Alum sludge with defined precautionary measures can be applied in both dry and crushed forms. Policymakers relevant to road construction can advise using this soil stabilizer to realize a low-cost stabilization, which can potentially save considerable expense during road construction projects. In the field, certain factors such as constant load application, changes to the water table, varied environmental conditions, and climatic changes over time may cause an increase or decrease in soil moisture content. Stabilized soil with alum sludge in such conditions strengthens the foundation of structures by beneficially changing the CBR of weak, clay-like soil versus soil with no alum sludge. Results indicate that an increasing percentage of alum sludge was at an optimum addition level of 8% [34]. As a low-cost solution, it can replace other stabilizers such as cement; it also offers a sustainable waste management solution.

Combinations of secondary raw material amendments are mentioned in multiple EU projects such as WAB (Wetlands, Algae, Biogas), RBR (Reviving Baltic Resilience), CONTRA (Baltic Beach Wrack—Conversion of a Nuisance to a Resource), and COST (European Cooperation in Science and Technology) actions like MINEA (Mining the European Anthroposphere), Marine Biotechnology, Protection, Resilience, and Rehabilitation of Degraded Environment to implement synergies and a multidisciplinary approach by combining two waste streams, organic and industrial, in the creation of added value products that will be proposed through multiple stakeholder channels for business and municipalities. Improving the system dynamics for waste streams and fixing unresolved issues on secondary raw materials from municipal activities and energetics are frontier questions for life cycle improvement of primary products, setting an advanced future for extended producers' responsibility.

## 5. Conclusions

The geotechnical engineering community is forced to become more sustainable in choosing amendments and materials used in the construction industry for weak soil stabilization, including peatlands. Conventionally manufactured materials are currently intended to be replaced by secondary raw (waste) material to achieve long-term sustainability goals.

A growing global economy demands good infrastructure covering larger areas; however, economic expansion must be balanced with environmental concerns and circular economy aspects. Areas covered worldwide by geotechnically challenging substrates such as peat demand a tremendously high amount of resources if road infrastructure is to be promoted and elaborated; therefore, industrial waste materials and byproducts available as secondary resources are essential, especially in light of potential shortages of resources in future. These incentives are combined with worldwide policies such as 'Green Deal' and 'EU Industrial Strategy' and innovation in the circular economy from academic and applied perspectives. The use of secondary raw materials is significant for the reduction of waste amounts as well as lowering the dependence of supply chains of raw materials for infrastructure building.

After a critical review of existing scientific literature, the general recommendation indicated the use of secondary raw materials as alternatives to conventional materials in peatland stabilization depending on regional and local industry features, historical storages of secondary waste (e.g., billions of tons of oil shale ash in the case of Estonia), and environmental and social policies. Implementation and application should be approved after careful environmental impact assessments such as evaluation of leaching, geotechnical bearing capacities, etc. In this way, a considerable amount of energy and resources will be saved, boosting the innovative recycling industries and creating a sustainable future.

**Author Contributions:** Conceptualization, Z.V.-G., T.T. (Tonis Teppand), and J.B.; methodology, T.T. (Tonis Teppand), J.B., M.K. (Mait Kriipsalu), M.K. (Maris Krievans), and Y.J.; validation, Z.V.-G. and M.K. (Maris Klavins); formal analysis, J.B., E.S., and I.Z.; investigation, T.T. (Tonis Teppand), M.K. (Mait Kriipsalu), and Y.J.; resources, M.K. (Maris Klavins) and T.T. (Toomas Tamm); data curation, Z.V.-G., I.G., V.R., and T.T. (Toomas Tamm); writing—original draft preparation, Z.V.-G., T.T. (Tonis Teppand), J.B., I.G., and V.R.; writing—review and editing, Z.V.-G., R.H.S., T.T. (Toomas Tamm), and M.S.; visualization, Z.V.-G., J.B., and M.K. (Maris Krievans); supervision, Z.V-G.; project administration, Z.V.-G.; funding acquisition, Z.V.-G. All authors have read and agreed to the published version of the manuscript.

**Funding:** This study was supported by project No. 1.1.1.2/VIAA/1/16/029 "Formula of peat-free soil conditioner with controlled-release fertilizing effect applicable for soil remediation and quality improvement of agricultural production" funded by ERDF.

**Data Availability Statement:** Not applicable.

**Acknowledgments:** The paper was elaborated in cooperation with the project No. 1.1.1.2/VIAA/ 1/16/029 "Formula of peat-free soil conditioner with controlled-release fertilizing effect applicable for soil remediation and quality improvement of agricultural production" carried out at the University of Latvia, projects No. KIK-15401 "Humiinaineid sisaldava pinnase stabiliseerimine teemulletes põlevkivituha abil / Stabilization of soil containing humic substances in roads with the help of oil shale ash" and No. SLTKT20427 carried out at the Estonian University of Life Sciences and project SARASWATI 2.0 carried out at the University of Tartu.

**Conflicts of Interest:** The authors declare no conflict of interest. The funders had no role in the design of the study; in the collection and interpretation of data; in the writing of the manuscript, and in the decision to publish the paper.

## Appendix A

**Table A1.** Recently studied materials applicable for weak soil stabilization: pros and cons.

| Type of material | Advantages | Disadvantages | Reference |
|---|---|---|---|
| 1. Conventional raw materials | | | |
| 1.1. Primary raw materials (industrial products) | | | |
| Cement | + Broad availability at market <br> + Wide applicability | – Large amount needed results in high costs <br> – Slow pozzolanic reaction <br> – Brittle behavior (may be affected by seismic activities) <br> – Relatively high carbon footprint <br> – Affects occupational health safety | [35,211] |
| Lime | + The oldest material used for stabilization with broad availability <br> + Wide applicability due to various states: quicklime, hydrated lime, or liquid lime | – Quarrying resource | [35,44] |
| 1.2. Conventional secondary raw materials | | | |
| Fly ash of class C | + Self-cementing (pozzolanic) properties <br> + Applicable as a component in the production of cement and cement clinker | – May induce environmental problems, mainly groundwater contamination, due to leaching of trace elements | [88,89,125,138,155,161,162,212] |
| 2. New choice of secondary raw materials | | | |

**Table A1.** *Cont.*

| Type of material | Advantages | Disadvantages | Reference |
|---|---|---|---|
| 2.1. Thermally treated waste products (ashes) | | | |
| 2.1.1. Ashes from agricultural production | | | |
| Bagasse ash | + Contains a significant amount of silica, thus, is considered as a sensible pozzolanic material with non-reactive behavior<br>+ Has a potential to be used in road subgrade stabilization | – Limited regional availability (in tropical areas only) | [213–216] |
| Rice husk ash | + Ensures improved sulfate resistance by stabilization of C-S-H and the refinement of pore structure embodied $CO_2$ emission and energy consumption | – Not a self-cementitious material, if used in stabilization, needs the addition of a hydraulic binder such as lime<br>– Limited regional availability (in tropical areas only) | [216–218] |
| 2.1.2. Ashes from energy production | | | |
| Fly ash | + Improves compressive and shearing strength of soils<br>+ Applicable as a stabilizing agent due to its siliceous and calcareous character | – May contain hazardous compounds released by leachate | [128,155,156,159–162,164–167,172,219] |
| Fuel oil fly ash | + Applicable as a stabilizing amendment and for adding to mortars | – May contain hazardous compounds released by leachate | [220] |
| Coal ash | + Wide applicability, including use as a top layer on unpaved roads | – May induce environmental problems, mainly groundwater contamination, due to leaching of trace elements | [221] |
| Granulated blast furnace slag | + Applicable as a stabilizing agent due to siliceous and calcareous character | – Limited availability<br>– Leaching control needed<br>– Legislative gaps | [54,222–224] |
| Oil shale ash | + Applicable as a stabilizer/binder for road/railway construction in unstable or contaminated soils<br>+ Applicable to create low carbon solutions and as a replacement of cement | – Limited availability<br>– Leaching control needed<br>– Legislative gaps | [101,107,110,115,116,118,119,195,196] |
| Semi-coke | + Applicable as a stabilizer/binder for road/railway construction in unstable or contaminated soils | – By itself has a high toxicity level<br>– Leaching control needed<br>– Legislative gaps | [100,108–113] |
| Palm oil fuel ash (POFA) | + POFA-peat composites ensure increased CBR values by >20% in comparison to untreated peat | – Limited regional availability (in tropical areas only) | [185–187] |
| 2.1.3. Ashes from various manufacturing | | | |
| Cement kiln dust or limestone ash (waste from cement and cement clinker production) | + Highly alkaline (contains 70% CaO and 30% undecomposed $CaCO_3$), thus, applicable as a substitute for lime | – May contain hazardous compounds released by leachate<br>– Harmful to the environment due to caustic nature | [44,212,225–227] |
| Sawdust ash (waste from woodworking) | + Mainly consists of silica, in the presence of moisture forms cementitious compounds<br>+ Improves strength and compressibility properties of soils<br>+ Due to low energy consumption and carbon footprint of activated fly ash, applicable as a replacement for cement | – By itself has a little cementitious value | [174,176,228] |
| Ashes from pulp and paper industry | + High moisture-holding capacity due to hydrophilic nature<br>+ Consists mainly of silica, aluminum oxide, and iron oxide, but it depends on wood species<br>+ Exhibits high acid-neutralizing capacity, thus, inducing a liming effect<br>+ Due to low energy consumption and carbon footprint, presence of reactive silicates and calcium carbonate, applicable as a replacement for cement | – High content of sulfates may affect the application | [171,173,177,178] |
| 2.1.4. Ashes from waste processing | | | |
| Sewage sludge ash | + If used in combination with cement, improves the California Bearing Ration (CBR) values by up to 30 times in comparison to untreated soil | – Toxicity risks, including contamination with pharmaceuticals and their residues<br>– Pharmaceutical pollutant risks | [180,183,184,229,230] |
| Municipal solid waste (MSW) incineration products (fly ash and bottom ash) | + Strengthens geotechnical parameters of soil in combination with a binder material<br>+ Bottom ash is applicable as an aggregate material | – May contain hazardous compounds released by leachate | [136–142] |

**Table A1.** *Cont.*

| Type of material | Advantages | Disadvantages | Reference |
|---|---|---|---|
| 2.1.5. High carbon content pyrolysis products | | | |
| Pyrogenic carbonaceous materials (PCM) and biochar | + Applicable for replacement of cement, but in small amount (<1%) or carbon sequestering admixture in cement<br>+ Useful for contaminated/polluted soils | – Variable composition and properties<br>– Limited amount can be added<br>– Mixing with other materials is needed | [231,232] |
| 2.2. Untreated waste and new products made from secondary raw materials | | | |
| 2.2.1. Waste from municipal biological treatment (MBTs) rejected materials and landfills | | | |
| Waste sludge, alum sludge | + Consists mainly of aluminum oxide (>30%), if used for soil stabilization, may significantly increase the CBR values<br>+ Has good pozzolanic properties<br>+ Small optimum amount (~8%) to be added | – Energy consumption, if drying is needed prior application | [34,209,210] |
| Fine fraction of landfilled waste | + Closing the loop of the material cycle in the circular economy<br>+ Methane degradation potential | – Toxicity problem<br>– Non-homogenous material<br>– Legislative gaps | [30,233–237] |
| 2.2.2. Waste form various industries | | | |
| Lignosulfonates (waste from wood pulping industry | + Chemical and physical binding of soil particles<br>+ Smaller amount is needed than conventional materials resulting in cost-efficiency<br>+ Non-brittle behavior (unaffected by seismic activities) | – Limited availability | [5,35,202] |
| Natural plant fibers (agricultural waste) | + Easy accessible material<br>+ Large amounts available | – Non-homogenous material<br>– Legislative gaps | [203–206] |
| 2.3. New products made from secondary raw materials | | | |
| 2.3.1. Composite materials | | | |
| Geopolymer composites | + Similar effect as using conventional materials<br>+ Smaller amount is needed than conventional materials (addition to soil 5–30% by weight) resulting in cost-efficiency | – Mixing with an alkaline activator is required | [87,89–92,96,175,178,195,198,199,201,222] |

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
