# Peer review of "Towards Sustainable Soil Stabilization in Peatlands: Secondary Raw Materials as an Alternative"

_sustainability, doi:10.3390/su13126726_

Round 1

Reviewer 1 Report

The topic of the manuscript is interesting for the audience. However, the structure of the manuscript is peculiar. The title is related to circular economy approach for soil stabilization in Peatlands, but the contents seem more related to the technical aspect of soil stabilization. The manuscript only presents the review of different literatures on soil stabilization and a little bit on circular economy. There is no originality of the manuscript. There is a need to have additional analyses with respect to circular economy or technical aspects of soil stabilization if the manuscript wants to go for possible publication. Additional comments are shown below:

  1. The abstract should be revised to include a clear objective, research methodology, analyses and conclusion.
  2. The English writing needs to improve. Major issues on grammar and vocabulary in the manuscript.
  3. Only review of literatures without in depth analyses can not be accepted as scientific publication especially in first tier journal like Sustainability.

Author Response

Dear Reviewer, we appreciate your time spent for reviewing our manuscript. We have discussed the recommendations in a proper way to improve the quality of the manuscript. The title was changed and specified to avoid the emphasis on the circular economy, as well as the aim. However, we did not go into a very detailed analysis because it can be a topic for other articles. We tried to stay within the frame of review to discuss the opportunities for secondary raw materials to be applied in geotechnical peatland stabilization.

Additional comments are shown below:

1. The abstract should be revised to include a clear objective, research methodology, analyses and conclusion.

Yes, indeed, the abstract generally should contain stated paragraphs, but in the case of a review paper, it is not compulsory. As indicated in the author’s instruction for ‘Sustainability’, review manuscripts should comprise the front matter, literature review sections and the back matter, and it is not necessary to follow the remaining structure.

2. The English writing needs to improve. Major issues on grammar and vocabulary in the manuscript.

Unfortunately, we cannot agree with this statement as a native American co-author edited the manuscript. In addition, a proofreading tool was applied to avoid grammar and style mistakes. Furthermore, other reviewers have indicated that English is good or minor English editing is needed.

3. Only review of literatures without in depth analyses can not be accepted as scientific publication especially in first tier journal like Sustainability.

We cannot fully agree with this statement. According to Palmatier et al. (2018), ‘in general, review papers are critical evaluations of material that has already been published performing a careful analysis of relevant literature to evaluate a specific research question, substantive domain, theoretical approach, or methodology and thereby provide readers with a state-of-the-art understanding of the research topic’. Why do we think that this paper is valuable for ‘Sustainability’? – we have tried to go through a deeper analysis of specific topic such as soil stabilization from broader horizons such as global sustainability aims without addressing the topic for a narrow niche of interests. (Palmatier W., Houston M.B., Hulland, J. 2018. Review articles: Purpose, process, and structure. J. Acad. Mark. Sci., 46, 1–5.)

Reviewer 2 Report

The authors make a comprehensive review of the valorization of secondary raw materials in geotechnical stabilization of peatlands from recent scientific published studies. Both the title, abstract, figures, and structure are adequate. I believe that this is a relevant contribution to an issue of great interest in engineering applications that deserves publication.

Author Response

Dear Reviewer, we kindly appreciate your positive evaluation of our manuscript.

Reviewer 3 Report

An impressive list of references!

I have attached a detailed list of comments. Here are some aspects I would like you to highlight or take into account:

Stablisitation is seldom superior to other methods but has a great potential to gain market share if the costs were lower. The problem is the requirement of materials not present on-site, expensive equipment and production rate. Be more fair to the other methods and focus on the potential of stablisation and the need for development needs in the area. 

The costs are important. Who is going to pay - is it the waste producer or the infrastructure project. Transportation of materials to site and the availability of the binders is usually the challange. The topic is valorazation of potential binder materials and it is not discussed. 

Peat is actually pretty good to contruct on. It is foundations for houses and similar that is most problematic to solve without e.g. stabilisation. 

A construction projects environmental impact is commonly governed by the access of on-site materials. Transportation of slag and ashes over longer distances ruins the greenhouse and energy budgets.

A discussion section would be great to sum up the findings, putting them into a perspective and outline for the conclusions. 

The scope in the introduction doesn't completely match the abstract. The valorization is missing

Author Response

Dear Reviewer, thank you for the remarks and comments; detailed answers are given below.

- Stablisitation is seldom superior to other methods but has a great potential to gain market share if the costs were lower. The problem is the requirement of materials not present on-site, expensive equipment and production rate. Be more fair to the other methods and focus on the potential of stablisation and the need for development needs in the area.

In several ways, it is mentioned as remote areas need solutions for new roads, plus have industry residuals that may serve as stabilization agent. If taken complete ecosystem and life cycle estimation, then many approaches would become favourable, e.g., 1) tax change, 2) wastegate fees, 3) policy change, 4) geographical specifics, etc. – it can be another manuscript. No one in forestry will make pile embankments if an easy approach is accessible, plus waste ash mountain is nearby like indicated in the Estonian case.

- The costs are important. Who is going to pay - is it the waste producer or the infrastructure project. Transportation of materials to site and the availability of the binders is usually the challange. The topic is valorazation of potential binder materials and it is not discussed.

Costs are dependent on the calculation method. If one changes the rules: tax, policy, strategy – the costs will be dependent on that. Who cared about wastewater environmental damage cost in older times? Why cared to clean it up? It is a policy change that also determines the emissions, damages, raw material substitution.

- Peat is actually pretty good to contruct on. It is foundations for houses and similar that is most problematic to solve without e.g. stabilisation. 

It is possible to build on piles, but no constructor will design a house or road without extracting away the peat or stabilizing it. Geotechnical properties are unreliable, and built premises will subside at various rates. The worst will happen in various humidity and or permafrost/seasonal frost regimes. Take a Northern Siberia railway project in Stalin times – it entirely deteriorates as the peat was not considered. Of course, it is possible to dry around the land in Europe and call the peat, which is not usual peat, any more peat. However, here in our manuscript, we consider the needs for remote regions and requirements for local roads and forestry use. The railway is different versus the countryside road in forest peatland.

- A construction projects environmental impact is commonly governed by the access to on-site materials. Transportation of slag and ashes over longer distances ruins the greenhouse and energy budgets.

The paper does not insist on always using only stabilization with secondary raw material. It is just an option which, in favourable circumstances, is better and expected from the circular economy point of view rather than expensive highway-type geotechnical works like in Europe where natural conditions are almost only in nature parks. For example, in Buryatia Sayani mountains, people travel on horses or by feet and crossing the peatland is impossible with four-wheel drives. Most of the heating nearby is performed by coal burning. Thus the stabilization material is available.

- A discussion section would be great to sum up the findings, putting them into a perspective and outline for the conclusions.

The review paper considers various stabilization technologies as an offer to implement circular economy principles in peatland stabilization; however, of course, there are many other ways to build on peatlands, extract peat away etc. Discussion is throughout the paper.

- The scope in the introduction doesn't completely match the abstract. The valorization is missing.

The title of the paper is changed as well as the aim is specified.

Reviewer 3, additional comments (from file)

General comments:

  • The discussion section is missing

Discussion is throughout the paper, and if needed, in each reference, the reader may get to the original paper to see the discussions on each aspect separately.

  • The environmental impacts on the peatland due to stabilisation is missing, e.g., barrier effects, changed hydraulic conditions

The paper does not insist to stabilize peatlands for highways but for basic infrastructure needs as the forestry cars or SUVs may cross the countryside road in any condition as in spring it is not possible.

  • The construction options to stabilization is poorly described. Floating roads and pre-loading utilises material on-site and will always be environmental effective. Stabilization is used when these options doesn’t working practice

It is disputable; however, if the secondary raw material is not far located and is subject to reuse anyway by waste management strategies, then the Green Purchase will choose waste valorization instead of using the primary raw material.

  • Since the topic is valorization of secondary raw material the aspect of the market conditions is missing: How cost effective is it to replace binders, how do meet the project requirements in volumes etc.

The scope of the paper is limited, and we were not analyzing all possible aspects, just selected ones.

  • The advantages of stabilization is the flexibility in treating deposits by depth and the combination of settlement reduction and strength increase. But it is costly and always requires transport of binders etc. that are not available on-site.

The purpose of the paper is to provide information on various stabilization options by secondary raw materials. We are not aimed to provide a technical economic analysis report here.

  • Peat properties is poorly described and underestimated from an engineering perspective. Heavy roads and railways can be constructed on peat but not foundations.

See the answer to the following comment.

Comments indicated by line numbers:

29 Several options are missing such as stepwise pre-loading, floating roads based on geogrid reinforcement and piled embankments. Both options doesn’t require any excavation or replacement of materials.

This paper is focussing on secondary raw material as an alternative opportunity for stabilization. The point is not to describe all possible geotechnical approaches. However, in a wide range of geography, there is much secondary waste that is never used and the need to build roads in remote regions always exists. Geogrids and piled embankments require additional raw material, but waste is waste anyway – it is an ecosystem calculation benefit if one uses waste instead of raw produced material. It is what we are discussing about.

30 Stabilization is an option if pre-loading or floating road is not favourable.

See the answers to previous and following comments.

33 Stabilization can never compete with pre-loading or floating roads in greenhouse gas emissions or energy consumption.

It is common practice that new approaches seem unsuitable for routine solutions at the first moment. For example, reverse osmosis in wastewater treatment was just a dream in the 90-ties because it was too expensive. Nevertheless, experience comes with time and sustainability issues insist looking into the future for alternative methods, materials, approaches, technologies. Various conditions always exist where one or another way of calculation makes things compatible and competitive. Costs are dependent on the calculation method. If one changes the rules: tax, policy, strategy – the costs will be dependent on that. Who cared about wastewater environmental damage cost in older times? Why cared to clean it up? It is policy change that also determines the emissions, damages, raw material substitution.

58 Substitute related facts by complementary facts.

Done.

77 In the sub-arctic areas we also have seasonal frost variations, not only permafrost.

Done.

79 The sentence includes too many statements. Split into two different: 1) There is a need to develop methods, 2) For soil stabilization the need of development is…

Done.

84 Typo: Remove “(“

Done.

98 Missing the aim of the article in the abstract.

The aim is indicated in the introduction. The abstract is limited to about 200 words, and we included the main ideas.

107 This statement is wrong – it is not impossible. The range of options are many, e.g., pilling, geogrid reinforcement, replacement, pre-loading… We are doing this on a daily basis.

It is always a question of policy and costs. No one will allow (and there is no need) in forestry roads build with mentioned methods due to costs and necessity estimations. The paper does not insist on always using only stabilization with secondary raw materials. It is just an option which in favourable circumstances is better and expected from a circular economy point of view rather than expensive highway-type geotechnical works like in Europe where natural conditions are only in nature parks. For example, in Buryatia Sayani mountains, people travel on horses or by feet and crossing the peatland is impossible with four-wheel drives. Most of the heating nearby is performed by coal-burning; thus, the stabilization material is available. Moreover, this is not a question of costs but of the possibility to build basic infrastructure at all.

117 Compression (loss of water) increases the strength and reduces the compressibility of peat. We have the highest axle loads on freight traffic in railways in Europe running over compressed peat for a hundred years. It is if the peat is dried, i.e., the wetland conditions are destroyed the problem arises.

It is possible to build on piles, but no constructor will design a house or road without extracting away the peat or stabilizing it. Geotechnical properties are unreliable, and built premises will subside at various rates. The worst will happen in various humidity and or permafrost/seasonal frost regimes. Take a Northern Siberia railway project in Stalin times – it entirely deteriorates as the peat was not considered. Of course, it is possible to dry around the land in Europe and call the peat, which is not usual peat anymore, peat. However, here in the article, we consider the needs for remote regions and requirements for local roads and forestry use. The railway is different from the countryside road in forest peatland.

140 Piling, especially wood and concrete piles, are missing. Soil nailing doesn’t work in peat at all.

The paper is not focussing on mastered binders from common ingredients. The figure reveals a wider method description – for general soil stabilization, not only for peatlands to give the reader an overall view.

144 Add embankment piling.

Done.

168 In permafrost regions often the uppermost layer, the active zone, is subjected to freeze and thaw. The permafrost is below the active layer.

Done.

177 Remove “any other geotechnical improvement”, it is only valid for soil stabilization.

Done.

181 Figure 3b is only valid for soil stabilization and not for the “mechanical methods”.

The figure gives the overlook on methods and management of permafrost.

242 Table S1 in appendix.

Done.

242 “Regarding the…” This sentence is too long and uses too many “,”. Split.

Done.

249 Split the sentence at “,”. Two different messages in the sentence.

Done.

426 We don’t pave with gravel – it is an unbound wearing course.

More than 70% of roads in Russia and Eastern Europe are built from glaciofluvial sediments – sand and gravel with no additional asphalt covering.

Reviewer 4 Report

The paper presents an extensive and pertinent review on secondary raw (waste) materials replacing conventionally manufactured materials used in the construction industry for weak soil stabilization, including peatlands, to achieve long-term goals of sustainability.

The authors provide a good overview of current peatland stabilization. However, since they don't integrate their findings, they don't give much new information. They seem to be knowledgeable about peatland stabilization, but they don't apply the lessons learned from the literature to their research.

I appreciate the work done, but the authors need to invest more in a proper analysis of their findings following the chapters which the journal indicates. It can be worth the effort because research studies always have something interesting to offer.

I would suggest the authors to reorganize the work by adding the missing parts. Then, they will show how they use the literature in their research and, more importantly, define the gaps and what has the literature to gain from their findings.

English is good, however, some improvements would be needed.
I propose the following changes regarding the Introduction:

29 and the environmentally

31 a single replacement

34-35 Besides traditional material usage, soil stabilization is achievable through various secondary raw materials (listed according to their groups and subgroups)

41 the excessive amounts of waste accumulation

42 in the circular economy

48-52 Peatlands cover more than 4 million km2 of the earth’s area primary providing such globally significant ecological services and functions such as water storage and filtration, flood control, coal (peat) management, carbon storage and sequestration, transpiration cooling, habitat for wildlife, recreation area. They and generally involve bogs, swamps, marshes, mires and wetlands [1-5].  

52-53 Even though Although the most considerable areas are concentrated in the northern regions (Figure 1), peatlands are widespread from sub-arctic to equatorial zones.

53-54 but Yet, not all the peatlands may be yet precisely recognizable

63 peatlands. For

64 such cities such as Saint Petersburg and Amsterdam and airports of such as

65 are built sit on peatlands

66 make instead of makes

70 At an industrial scale, […] for fossil resource extraction

83-84 the integration of the circular economy approach (.

102 the circular

Author Response

Dear Reviewer, thank you for the comments and remarks addressed to improve our manuscript. As indicated in the author’s instruction for ‘Sustainability’, review manuscripts should comprise the front matter, literature review sections and the back matter, and it is not necessary to follow the remaining structure. Discussion is throughout the paper.

I propose the following changes regarding the Introduction:

29 and the environmentally

Done.

31 a single replacement

Done.

34-35 Besides traditional material usage, soil stabilization is achievable through various secondary raw materials (listed according to their groups and subgroups)

Done.

41 the excessive amounts of waste accumulation

Done.

42 in the circular economy

Done.

48-52 Peatlands cover more than 4 million km2 of the earth’s area primary providing such globally significant ecological services and functions such as water storage and filtration, flood control, coal (peat) management, carbon storage and sequestration, transpiration cooling, habitat for wildlife, recreation area. They and generally involve bogs, swamps, marshes, mires and wetlands [1-5].

Done. 

52-53 Even though Although the most considerable areas are concentrated in the northern regions (Figure 1), peatlands are widespread from sub-arctic to equatorial zones.

Done.

53-54 but Yet, not all the peatlands may be yet precisely recognizable

Done.

63 peatlands. For

Done.

64 such cities such as Saint Petersburg and Amsterdam and airports of such as

Done.

65 are built sit on peatlands

Done.

66 make instead of makes

Done.

70 At an industrial scale, […] for fossil resource extraction

Done.

83-84 the integration of the circular economy approach (.

Done.

102 the circular

Done.

Round 2

Reviewer 1 Report

The Authors have done some revisions with respect to English language. However, the response to the previous comment of no 3 is not acceptable. First, the citation that was used by the Author is from the other discipline (marketing). Second, even I can accept this reference, the Authors didn't put it as reference in the original and revised manuscript. It means the Authors didn't consider this reference from beginning of the preparation of the manuscript. The Authors only want to find excuse to defend the manuscript. I can accept the citation from the other discipline that the Authors used to prepare the manuscript, but the Authors must follow the latest citation. Please refer to the following citation for the Authors to revise the manuscript. The following citation has a good methodology to prepare the manuscript as review paper.

Hannah Snyder,
Literature review as a research methodology: An overview and guidelines,
Journal of Business Research, Volume 104, 2019, Pages 333-339, ISSN 0148-2963, https://doi.org/10.1016/j.jbusres.2019.07.039.

Author Response

We appreciate your ardour in discussing the issue regarding the structure of a review paper, and we do not mind continuing this discussion outside the frames of the current manuscript’s submission. If we judge strictly, then both of these mentioned articles are not entirely suitable for the current research area, which is basically environmental science with a specific direction to sustainable material recycling/reuse, and therefore will not be added to the list of references. In our opinion in this case, first of all, we have to take into account the requirements of the journal where the manuscript is submitted. In the section ‘Instruction for Authors’, when preparing the article for ‘Sustainability’ at MDPI, it is stated that ‘Review manuscripts should comprise the front matter, literature review sections and the back matter. [..] It is not necessary to follow the remaining structure’. Furthermore, our paper is not thought to be a meta-analysis; therefore, we believe that the structure can be developed considering the specifics of the reviewed data. By the way, several authors of this particular manuscript have also been involved in the elaboration of other review articles; one of them (Burlakovs et al., 2017) has enough high citing index (40 at Scopus on 2021-06-03) to be assessed as scientifically recognizable at the international level. We believe that a different than usual manner in writing scientific review papers may also serve as an innovative scientific approach broadening the viewpoint on the problem or topic raised. (Reference: Burlakovs, J., Kriipsalu, M., Klavins, M., Bhatnagar, A., Vincevica-Gaile, Z., Stenis, J., Jani, Y., Mykhaylenko, V., Denafas, G., Turkadze, T., Hogland, M., Rudovica, V., Kaczala, F., Rosendal, R.M., Hogland, W. (2017) Paradigms on landfill mining: From dump site scavenging to ecosystem services revitalization. Resources, Conservation and Recycling, 123, 73–84. https://doi.org/10.1016/j.resconrec.2016.07.007)